# Bioprospection of the trichomonacidal activity of lipid extracts derived from marine macroalgae G*igartina skottsbergii*

Tallyson Nogueira Barbosa[1], Mara Thais de Oliveira Silva[1], Ângela Sena-Lopes[1], Frederico Schmitt Kremer[2], Fernanda Severo Sabedra Sousa[3], Fabiana Kommling Seixas[3], Tiago Veiras Collares[3], Cláudio Martin Pereira de Pereira[4], Sibele Borsuk[1]*

1 Laboratório de Biotecnologia Infecto-Parasitária, Centro de Desenvolvimento Tecnológico, Biotecnologia, UFPel, Pelotas, Rio Grande do Sul, Brasil, 2 Laboratório de Lipidômica e Bio-orgânica, Centro de Ciências Químicas, Farmacêuticas e de Alimentos, UFPel, Pelotas, Rio Grande do Sul, Brasil, 3 Laboratório de Bioinformática e Proteômica, Centro de Desenvolvimento Tecnológico, Biotecnologia, UFPel, Pelotas, Rio Grande do Sul, Brasil, 4 Laboratório de Biotecnologia do Câncer, Centro de Desenvolvimento Tecnológico, Biotecnologia, UFPel, Pelotas, Rio Grande do Sul, Brasil

* sibeleborsuk@gmail.com

**Data Availability Statement:** All relevant data are within the manuscript and its Supporting information file.

## Abstract

Algal extracts are sources of bioactive substances with applications in the development of novel alternative drugs against several diseases, including trichomoniasis sexually transmitted infection caused by *Trichomonas vaginalis*. Factors such as clinical failures and resistant strains limit the success of the existing drugs available for treating this disease. Therefore, searching for viable alternatives to these drugs is essential for the treatment of this disease. The present study was conducted for, *in vitro* and *in silico* characterization of extracts obtained from marine macroalgae *Gigartina skottsbergii* at stages gametophidic, cystocarpic, and tetrasporophidic. In addition, antiparasitic activity of these extracts against the ATCC 30236 isolate of *T. vaginalis*, their cytotoxicity, and gene expression of trophozoites after treatment were evaluated. The minimum inhibitory concentration and 50% inhibition concentration were determined for each extract. Results: *In vitro* analysis of the extracts' anti-*T. vaginalis* activity revealed an inhibitory effect of 100%, 89.61%, and 86.95% for *Gigartina skottsbergii* at stages gametophidic, cystocarpic, and tetrasporophidic, respectively, at 100 μg/mL. *In silico* analysis revealed the interactions between constituents of the extracts and enzymes from *T. vaginalis*, with significant free energy values obtained for the binding. None of the extract concentrations exhibited cytotoxic effects on VERO cell line compared to control, while cytotoxicity on HMVII vaginal epithelial cells line was observed at 100 μg/mL (30% inhibition). Gene expression analysis revealed differences in the expression profile of *T. vaginalis* enzymes between the extract-treated and control groups. According to these results, *Gigartina skottsbergii* extracts exhibited satisfactory antiparasitic activity.

**Funding:** This study was financed in part by the Coordenação de Aperfeiçoamento de Pessoa de Nível Superior – Brasil (CAPES) – Finance Code 001. The funders had no role in study design, data collection and analysis, decision to publish, or preparation of the manuscript.

**Competing interests:** The authors declare that they have no competing interests.

## Introduction

Products derived from natural sources have been applied to various fields related to human health since the beginning of civilization [1]. These natural products are useful in the treatment of different diseases and offer numerous advantages, such as convenient production of their active biomolecules and reduced side effects on human health [2].

The biomolecules derived from natural sources, in contrast to those synthesized synthetically, present a structural diversity that enables a broad range of biological activity, which renders these biomolecules more promising in the field of novel drug development [3]. Marine organisms, such as macroalgae, are good examples of such natural sources of active biomolecules that are useful in the development of novel drugs and related therapies [4].

One prime species among such marine organisms is *Gigartina skottsbergii*, a species of red macroalgae native to the subantarctic region [5]. This organism is a source of certain important bioactive molecules that are associated with the unique metabolic pathways of this organism [6]. These active biomolecules exhibit diverse activities, including anti-inflammatory, anti-tumor, anti-coagulant, antioxidant, and antimicrobial activity [7]. Therefore, this species is considered important taking into account its possible application in the field of human health and protection from various pathogens, such as parasites, that cause diseases in humans [8].

Trichomoniasis is a disease caused due to infection with the protozoan *Trichomonas vaginalis.* [9]. This species is a flagellated parasite that is considered responsible for the most widespread protozoan infections worldwide [10]. Infections caused by this parasite lead to serious complications related to the human genitourinary system, particularly in female, such as vaginitis, vulvar pain, pregnancy-related complications, and the risk of developing cervical and prostate cancer [11].

Metronidazole is the drug mainly used for the treatment of trichomoniasis [12]. Although efficient, the drug reportedly achieves lower treatment success, as evidenced by the clinical failures and adverse effects observed. The emergence of strains resistant to this drug has also contributed to its lower success in the treatment of trichomoniasis [13]. Therefore, searching for novel therapies that are safer and further effective in combating trichomoniasis is becoming increasingly important [14]. In this context, the present study was conducted to evaluate the antiparasitic activity of the *Gigartina skottsbergii* derived lipid extracts against *T. vaginalis*.

## Material and methods

### Macroalgae specimens

The collections were carried out through a project in partnership between the Federal University of Pelotas and the University of Magallanes in Chile. The project was approved by PGCI/CAPES, an international cooperation program in scientific research, with registration number 99999.002378/2015-09.

The *Gigartina skottsbergii* specimens, each at a different developmental stage (gametophytic, tetrasporophytic, or cystocarpic), were collected manually from the Punta Arenas region (Latitude: –53.1667, Longitude: –70.9333, 53°10′0″South, 70°55′60″West). The collected specimens were washed and subjected to morphological characterization for identification. Afterward, the specimens were freeze-dried and then submitted to spraying steps, followed by storage in dark plastic bags inside a desiccator to protect them from elements such as heat, light, and humidity until used for analyses. After that, the identification of the macroalgae species was carried out in the Herbarium of the Laboratory of Antarctic and Subantarctic Marine Ecosystems (LEMAS) of the University of Magalhães (UMAG), located in Punta Arenas, southern Chile.

## Extraction and derivatization

Fatty acids were extracted from specimens as described by Bligh and Dyer (B&D) (1959) [15]. Briefly, 20 mL of methanol, 10 mL of chloroform, 10 mL of an aqueous solution of sodium sulfate at 1.5% (w/v), and 1 g of specimen biomass were mixed, and the resulting solution was placed in an ambient environment under constant stirring for 30 min. Subsequently, the sample solutions were transferred to different conical Falcon tubes and centrifuged at 3000 rpm for 30 min. The lower organic phase was recovered and dried under reduced pressure. The procedure was repeated three times (n = 3).

The derivatization of fatty acids was performed according to the method described by MOSS & collaborators (1974) [16]. The extracted material was submitted to a reflux system at 100°C for 5 min along with a 0.5 M methanolic solution of sodium hydroxide. Subsequently, 5 mL of a 14% methanolic boron trifluoride solution were added to the above sample, and the reflux was maintained at 100°C for another 5 min. Afterward, the system was cooled by adding 3 mL of a saturated aqueous solution of sodium chloride and 20 mL of n-hexane. Finally, the solution was transferred to a separatory funnel to recover the upper organic phase, which was then filtered with anhydrous sodium sulfate and dried under reduced pressure. The procedure was repeated three times (n = 3).

## Instrumentation and quantification

After the extraction and derivatization of fatty acids, the samples were diluted in n-hexane and subjected to a gas chromatography flame ionization detector (GC-FID) model GC-2010 (Shimadzu, Kyoto, Japan) using an SP-2560 capillary column (100 m × 0.25 mm × 0.2 μm) from Supelco (Bellefonte, USA) and nitrogen as the carrier gas. The initial column temperature was 140°C, which was raised gradually at the increments of 4°C/min to 240°C. This temperature was then maintained for 10 min, creating a total duration of 40 min for each run. The injector was maintained at 260°C, and the injections were performed in split mode (1:100). The identification and quantification of fatty acids were performed through comparison with the FAME 37-Mix standard using the GC Solution software (Shimadzu, Kyoto, Japan). All standards and chemicals used in the present study were of analytical grade.

## *In silico* analysis

**Protein and ligand structures and molecular docking.** The protein structures of methionine gamma lyase from *T. vaginalis* (TvMGL), purine nucleoside phosphorylase from *T. vaginalis* (TvPNP), triophosphate isomerase from *T. vaginalis* (TvTPI), and lactate dehydrogenase from *T. vaginalis* (TvLDH) were acquired from the Protein Data Bank (PDB: 1E5E, 1Z36, 3QST, and 5A1T, respectively), while those of papain-like cysteine protease from *T. vaginalis* (TvCP2), thioredoxin reductase from *T. vaginalis* (TvTrxR), and cathepsin l-type cysteine protease from *T. vaginalis* (TvCPCAC1) were predicted using I-TASSER [2] based on their respective primary sequences that were obtained from the UniProt program (UniProt: Q27107, Q8IEV3, and Q6UEJ4, respectively).

The ligand structures were converted from SDF to PDB format using the OpenBabel program [17], the results of which were tabulated using the script "prepare_ligand4.py" from AutoDock Tools to generate the PDBQT files for the docking procedure [18].

The receiver in the PDB files was further analyzed using the COACH program [19] to identify the potential bid pockets, followed by processing using "prepare_receiver4".py from AutoDock Tools and then conversion to PDBQT. Processed binders and receptors were finally docked at the predicted binding sites using the AutoDock Vina program [20].

**Parasite culture conditions.** The *T. vaginalis* isolate 30236 was obtained from the American Type Culture Collection (ATCC). This strain was used in the present study as it is normally susceptible to metronidazole. The trophozoites were axenically cultivated in the trypticase-yeast extract-maltose (TYM) medium without agar (pH 6.0), which was supplemented with 10% sterile bovine serum (SBS). An incubation temperature of 37°C was selected for culture as reported in a previous study [21].

**Antiparasitic assay.** The antiparasitic assay was conducted using the culture of *T. vaginalis*, trophozoites, and the lipid extracts *Gigartina skottsbergii* stage gametophidic (GFG), *Gigartina skottsbergii* stage tetrasporophidic (GFT), and *Gigartina skottsbergii* stage cystocarpic (GFC), as described by Sena-lopes *et al.* (2017) [22]. The cultures that exhibited viability equal to or greater than 95%, as confirmed in the motility analysis, morphology examination, and exclusion test with trypan blue dye (0.4%) performed under an optical microscope at 400x magnification. The tests were performed on plates of 96-well microtiter (Cral®). All assays were performed independently in triplicate.

The determined minimum inhibitory concentration (MIC) and 50% inhibitory concentration (IC50) values of the extracts against *T. vaginalis* were verified using a previously described methodology [23]. The parasites were seeded at an initial density of $2.6 \times 10^5$ trophozoites/mL in TYM medium at a final concentration of 150 ul of *T. vaginalis* trophozoites/well, and then incubated in the presence of GFG, GFC and GFT lipid extracts previously diluted in dimethyl-sulfoxide (DMSO).

The concentrations of 100 μg/mL, 50 μg/mL, 25 μg/mL, 12.5 μg/mL, and 6.25 μg/mL of the lipid extracts were evaluated. Three controls were also established for the assay: the negative control with only trophozoites, the positive control containing 100 μM of MTZ (metronidazole; Sigma-Aldrich), and the control containing the vehicle used for the solubilization of lipid extracts (0.6% DMSO). The plates were incubated at 37°C in the presence of 5% $CO_2$ for 24 h.

After incubation, a solution containing an aliquot of trophozoites and trypan blue (0.4%) (1:1, v/v) was prepared and observed in a Neubauer chamber to determine the motility, morphology, and viability of the trophozoites and then accordingly calculate the MIC. The determined MIC values were confirmed using the parasite pellets treated with lipid extracts at the determined MIC concentration along with the controls. In this MIC confirmation assay, the samples were incubated at 37°C in tubes containing 1.5 mL of fresh TYM medium supplemented with SBS and antibiotics. Subsequently, counts were performed in a Neubauer chamber with both trophozoite preparation and trypan blue solution between 24 h and 96 h. GraphPad Prism 7.0 software was employed to calculate the IC50 values.

**Kinetic growth curve of *T. vaginalis*.** In order to obtain a further precise activity profile of the lipid extracts against *T. vaginalis*, a kinetic growth curve was generated. The 96-well plates were prepared as described above and incubated with the lipid extracts at the respective MIC concentrations against *T. vaginalis*. Growth analysis was performed at 1, 6, 12, 24, 48, 72, and 96 h using the trypan blue dye exclusion test (0.4%). All assays were performed independently in triplicate. The results were expressed as the percentage of viable trophozoites relative to the untreated parasites.

**Cytotoxicity assay.** The cytotoxicity assessment was performed as described by Navarrete-Vázquez et al. (2015) [24] using two cell lines—VERO cells and vaginal epithelial cells (HMVII). A concentration of $2.5 \times 10^4$ viable cells was incubated in the wells of a 96-well plate containing either DMEM (VERO) or RPMI-1640 (HMVII) medium (as required) enriched with 10% Fetal Bovine Serum (FBS) and 2% antibiotic-antimycotic. The plates were then cultured for 24 h at 37°C under the conditions of 5% $CO_2$ and 95% humidity. When 75% confluence was reached, the spent medium was replaced with the fresh one, followed by treatment of cells with different concentrations (100 μg/mL, 50 μg/mL, 25 μg/mL, 12.5 μg/mL, and 6.25 μg/

mL) of each of the lipid extracts (GFG, GFT, and GFC) for 24 h at 37˚C in a 5% $CO_2$ atmosphere.

Four controls were used in the assay: a negative control containing untreated cells, a control containing the vehicle for the solubilization of extracts (DMSO 0.6%), a control of 100 μM MTZ, and a positive control with only DMSO. After incubation, the medium was removed and 100 μL of 3-(4,5-dimethylthiazol-2-yl)-2,5-diphenyltetrazolium (MTT) bromide solution (5 mg/mL) was added to each well, followed by 3 h of incubation at 37˚C in a 5% $CO_2$ atmosphere.

Afterward, the medium was removed again, and 100 μL of DMSO was added to each well to solubilize the formazan crystals that had formed. The reduction of MTT to formazan, which is in direct proportion to the number of living cells, was examined in a microplate reader at 492 nm. The results were expressed as the percentage of viable cells relative to the untreated cells. All observations were validated in a minimum of three independent experiments.

**Gene expression analysis of *T. vaginalis*.** Total RNA from *T. vaginalis* trophozoites ($10^7$ cells/sample) was extracted using the Trizol Reagent (Invitrogen) by following the manufacturer's instructions. The cDNA was synthesized from 0.5 μg of the extracted total RNA using the high capacity kit (applied biosynthesis) according to the protocol provided by the manufacturer. The PCR mix (20 μL) contained 10 μL of SYBR Green PCR Master Mix (Applied Biosystems, UK), 300 nM of primers, 1 μL of synthesized cDNA, and RNase-DNase free water. The PCR conditions were: a cycle at 95˚C for 5 min followed by 40 cycles at 95˚C for 10 s and then at 60˚C for 30 s. The melting curve analysis was performed following the cycling protocol of 95˚C for 15 s, 55˚C for 15 s, and 95˚C for 15 s.

All real-time PCR experiments were conducted in the Stratagene Mx3005P real-time PCR system (Agilent Technologies, Santa Clara, CA, USA) using the following primers:Actin Forward 5' `TCACAGCTCTTGCTCCACCA` 3' and Reverse 5' `AAGCACTTGCGGTGAACGAT` 3' sequences (GenBank accession number: KF747377.1) [25]; Pyruvate-ferredoxin oxidoreductase A (PFOR A) Forward 5' `CGGCTACGGTATGTTCAAGG` 3' and Reverse 5' `TCCTTGTCCTGATCCCAAAC` 3', sequences (GenBank accession number: U16822), and Pyruvate-ferredoxin oxidoreductase B (PFORB) Forward 5' `CTGCAAGCTCCTTACACAGC` 3' and Reverse 5' `AAGAGGGAGTTAGCCCAAGC` 3', sequences (GenBank accession number: U16823); Malic enzyme D Forward 5' `CATCTGTTAGCCTCCCAGTCC` 3' and Reverse 5' `ACGAGCAGCTTGTTCATCCT` 3', sequences (GenBank accession number: U16839); and Hydrogenase Forward 5' `TGCACACGAAAGAAGGATGA` 3' and Reverse 5' `TCGCATGGTGTATCTGGTAA` 3' sequences (GenBank accession number: U19897) [26]. The actin gene was used as normalizer in all analyses.

## Statistical analysis

Statistical analysis was performed through one-way analysis of variance (ANOVA) using a significance threshold of $p < 0.05$. Tukey's test was conducted to identify the significant differences between the mean values obtained for different treatments (Software GraphPad Prism 5.0).

## Results

### Chemical constitution of the lipid extracts

The GC-FID technique was applied to detect the fatty acids constituting the lipid extracts evaluated in the present study. The list of the identified fatty acids, along with the corresponding percentages of these compounds in the extracts, is given in Table 1. The GFG extract comprised 50.29% saturated fatty acids, 15.30% monounsaturated fatty acids, and 34.41%

**Table 1. Percentage indices of the fatty acids constituting the lipid extracts from *Gigartina skottsbergii* identified using gas chromatography flame ionization detector (GC-FID).**

| Fatty acid | *Gigartina skottsbergii* concentration (%) | | |
|---|---|---|---|
| | **Gametophytic** | **Cystocarpic** | **Tetraesporophytic** |
| Lauric acid (12:0) | 1.07 | - | 0.07 |
| Myristic acid (14:0) | 5.52 | 6.53 | 5.51 |
| Myristoleic acid (C14:1) | 1.14 | - | - |
| Pentadecanoic acid (C15:0) | 1.40 | - | - |
| Palmitic acid (C16: 0) | 34.46 | 39.74 | 36.47 |
| Palmitoleic acid (C16: 1) | 2.56 | 4.45 | 1.86 |
| Heptadecanoic acid (C17: 0) | 1.65 | - | - |
| Stearic acid (C18: 0) | 6.19 | 13.65 | 4.77 |
| Oleic acid (C18: 1n9c) | 11.62 | 14.03 | 15.40 |
| Linoleic acid (C18: 2n6c) | 2.15 | 6.59 | 3.36 |
| cis-8,11,14-Eicosatrienoic acid (C20:3n6) | 2.22 | - | 1.29 |
| Arachidonic acid (C20:4n6) | 15.50 | 6.86 | 15.58 |
| Eicosapentaenoic acid (C20:5n3) | 14.56 | 8.15 | 15.08 |
| Nervonic acid (24:1n9) | - | - | 0.61 |
| Saturated | 50,29 | 59,92 | 46,82 |
| Monounsaturated | 15,30 | 18,48 | 17,87 |
| Polyunsaturated | 34,41 | 21,6 | 35,31 |

polyunsaturated fatty acids. The GFC extract comprised 59.92% saturated, 18.48% monounsaturated, and 21.6% polyunsaturated fatty acids. The GFT extract comprised 46.82%, 17.87%, and 35.31% of saturated, monounsaturated, and polyunsaturated fatty acids, respectively.

Therefore, a considerable variation of the constituent fatty acids was observed in the extracts, which was probably because the extracts were from different development stages of the same species of macroalgae (Table 1).

## Anti-*T. vaginalis* activity

The lipid extracts exhibited *T. vaginalis* trophozoite-inhibition to different degrees after 24 h of exposure at a concentration of 100 μg/mL. At their respective MICs, GFC and GFT reduced the viability of trophozoites by 89.61% and 86.95%, respectively, while GFG induced the complete inactivation of the parasite. The IC50 value determined for GFG and GFT was 50 μg/mL, while that for GFC was 25 μg/mL. As expected, the positive control with MTZ reduced the viability *T. vaginalis* by 100%, while the vehicle dilution control (0.6% DMSO) exhibited no significant difference from the negative control of untreated trophozoites (Fig 1).

The kinetic growth curve revealed that exposure to 100 μg/mL of GFG reduced the growth of trophozoites by 72% within 12 h, with no significant difference compared to the positive control of MTZ after 24 h of exposure. The exposure to 100 μg/mL of GFC and GFT reduced the trophozoite growth by 46% and 47%, respectively, within 12 h. Although 24 h of exposure to GFC and GFT did not cause a complete termination of trophozoite growth, persistent inhibition of trophozoite development was continued (Fig 2).

## Cytotoxicity assay

The *in vitro* cytotoxicity analysis of the extracts in the mammalian VERO cell line revealed cell viability of 100%, 98.72%, and 98.20%, respectively, at their respective MIC values after 24 h of

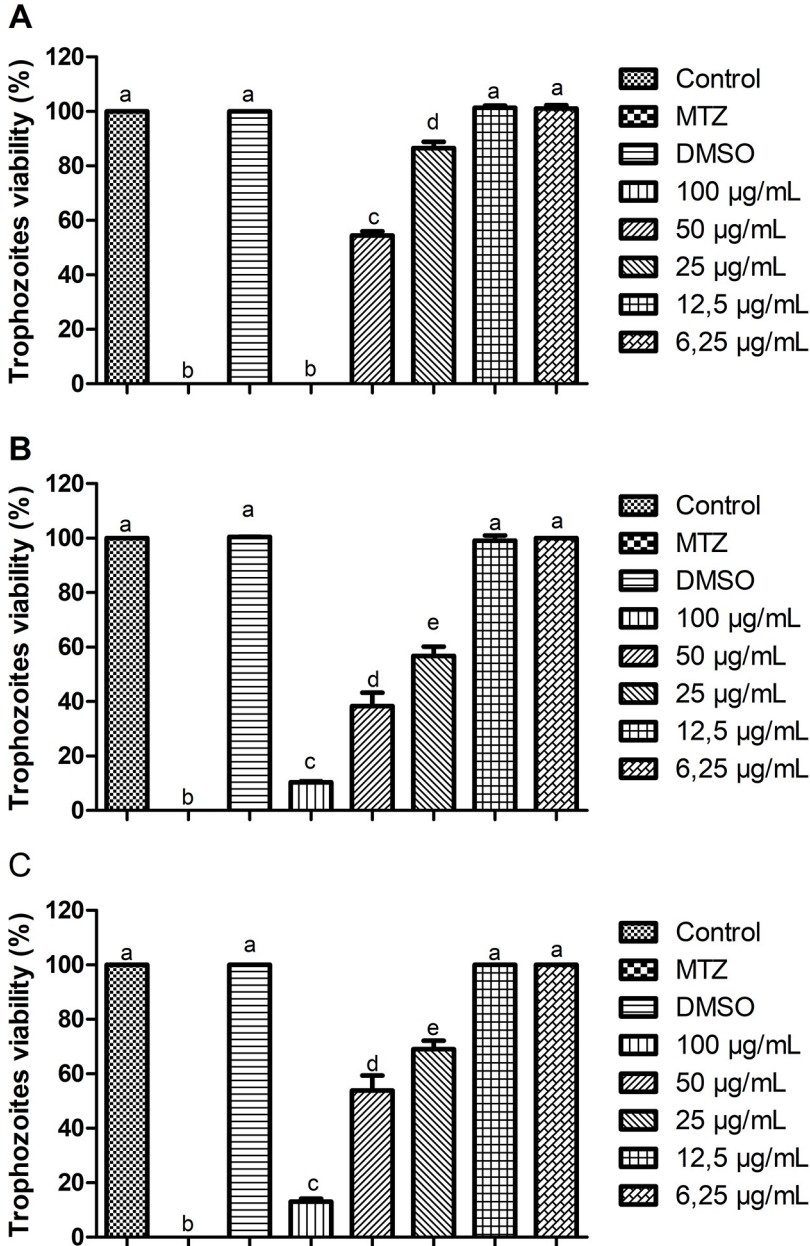

**Fig 1.** Determination of the MIC and IC50 values against the ATCC 30236 isolate of *T. vaginalis* after treatment with the lipid extracts (A) *Gigartina skottsbergii* stage gametophidic (GFG), (B) *Gigartina skottsbergii* stage cystocarpic (GFC) and *Gigartina skottsbergii* stage tetrasporophidic (GFT) at different concentrations (100, 50, 25, 12.5, and 6.25 μg/mL), after 24 h of exposure. Control (untreated trophozoites), DMSO (vehicle for solubilization), and MTZ (100 μM metronidazole). Data are presented as the mean ±standard deviation of a minimum of three experiments. Different letters indicate a significant difference (p < 0.05).

exposure. These results did not differ significantly from those obtained for the negative control of untreated cells (Fig 3).

When evaluating the cytotoxicity of the extracts in the HMVII cell line, cell viability of 77.32%, 80.64%, and 82.60% was observed for GFG, GFC, and GFT, respectively. In both cell lines subjected to cytotoxicity evaluations, the vehicle control (0.6% DMSO) did not alter cell growth, while the positive control (DMSO) caused a 100% reduction in cell viability (Fig 4).

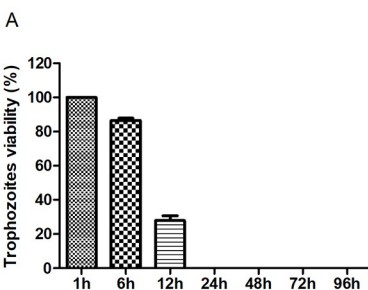 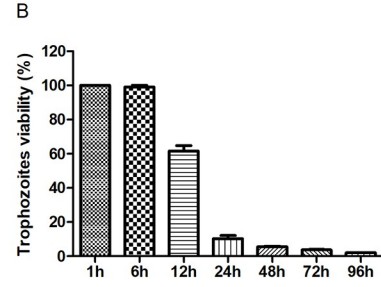 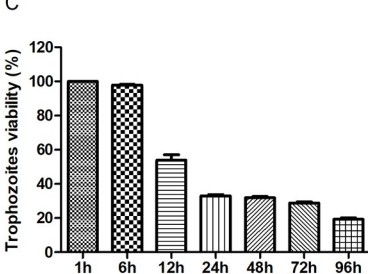

**Fig 2.** Kinetic growth curve of the ATCC 30236 isolate of *T. vaginalis* after treatment with the lipid extracts (A) GFG, (B) GFC, and (C) GFT, for the exposure durations of 1, 6, 12, 24, 48, 72, and 96 h. Data were presented as the mean ±standard deviation of a minimum of three experiments. Different letters indicate a significant difference (p < 0.05).

## Molecular docking

Molecular docking was conducted to analyze the interactions of the chemical constituents of the GFG, GFC, and GFT extracts with potential enzyme targets to ultimately evaluate the anti-*T. vaginalis* activity of the extracts.

The docking analysis revealed variable free energy values (Δ Gbinding) for each interaction analyzed. The best free energy of binding value of −6.7 kcal/mol was generated between arachidonic acid and TvLDH. The interaction between eicosapentaenoic acid and TvLDH, eicosapentaenoic acid and TvTrxR, and linoleic acid and TvTrxR also exhibited a considerable amount of free energy (−6.6 kcal/mol, −5.4 kcal/mol, and −5.6 kcal/mol, respectively).

The interactions with metronidazole, a reference drug used for the treatment of infections caused by *T. vaginalis*, were also analyzed, and the highest binding free energy scores were obtained in interaction with TvMGL (−6.5 kcal/mol) and TvLDH (−6 kcal)/mol). These values were similar to those obtained for the constituent compounds of the evaluated extracts (Table 2).

## Gene expression in *T. vaginalis* after treatment with GFG, GFC, and GFT

The expression levels of PFOR A, PFOR B, hydrogenase, and malic enzyme genes in *T. vaginalis* after the trophozoites were exposed to the GFG, GFC, and GFT extracts at MIC (100 μg/mL) were analyzed using qRT-PCR. Significant differences were observed in the expression of these genes between the extract-treated and control groups (untreated trophozoites, trophozoites treated with the solubilization vehicle, and trophozoites treated with the commercially available drug for the treatment of trichomoniasis-MTZ) (Fig 5).

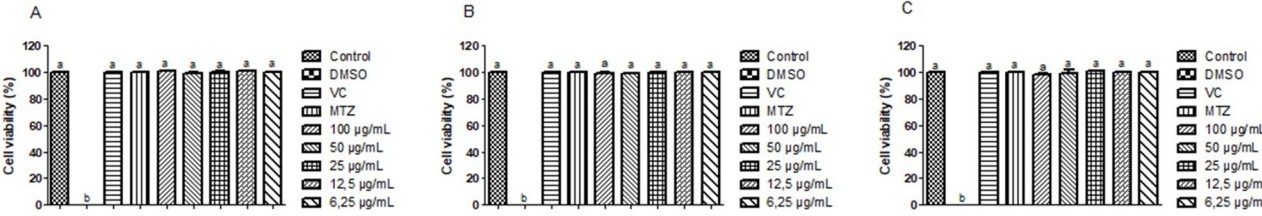

**Fig 3. VERO cell viability after treatment with different concentrations of the lipid extracts (A) GFG, (B) GFC, and (C) GFT.** Cell proliferation in VERO cells was investigated after 24 h of exposure using the MTT assay. Control (negative control—untreated cells), DMSO (positive control), VC (solubilization vehicle—DMSO 0.6%), and MTZ (metronidazole at 100 μM). Data are presented as the mean ±standard deviation of a minimum of three independent experiments. Different letters indicate a significant difference between the treatments (p < 0.05).

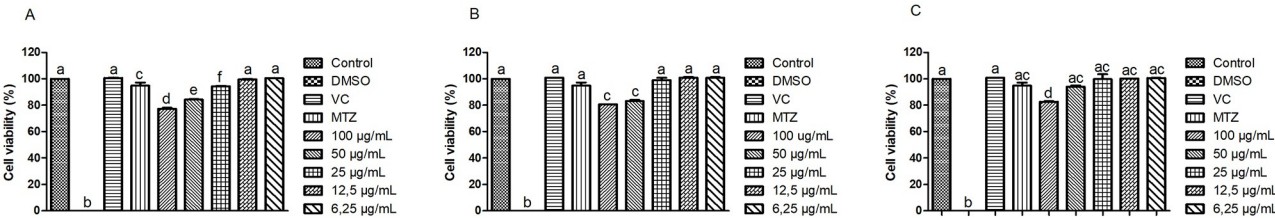

**Fig 4. Cytotoxicity effect on the vaginal epithelial HMVII cells after 24 h of exposure to different concentrations of the lipid extracts (A) GFG, (B) GFC, and (C) GFT.** Cell proliferation in HMVII cells was investigated using the MTT assay. Control (negative control—untreated cells), DMSO (positive control), VC (solubilization vehicle– 0.6% DMSO), and MTZ (metronidazole at 100 μM). Data are presented as the mean ±standard deviation of a minimum of three independent experiments. Different letters indicate a significant difference between the treatments ($p < 0.05$).

GFG was observed to promote an increase in the expression levels of malic enzyme D, hydrogenase, PFOR A, and PFOR B, while GFC promoted a significant increase in hydrogenase expression compared to the control group and a reduction in the expressions of PFOR A and PFOR B. GFT exhibited a significant increase in the expression of the malic D enzyme compared to the control groups, although the levels were not statistically different from those observed for GFC. In addition, GFT promoted a significant increase in the expression of PFOR B compared to the other control groups, GFG, GFC, and MTZ, while inhibiting the expression of PFOR A compared to these other treatments. No significant differences were observed in the gene expression between the control group and the MTZ group.

## Discussion

Seaweeds are considered true biological sources of natural bio-actives that have applications in different fields [27]. The present study described the use of lipid extracts derived from subant-arctic macroalgae *Gigartina skottsbergii* as an antiparasitic against *T. vaginalis* and the profile

**Table 2. Free energy levels (kcal/mol) determined in the molecular docking analysis for the interactions of the enzymes of interest involved in the metabolism of *T. vaginalis* with the fatty acids constituents of the lipid extracts from *Gigartina skottsbergii*: Papain-like protease (TvCP2), triosephosphate isomerase (TvTPI), lactate dehydrogenase (TvLDH), methionine gamma ligase (TvMGL), cathepsin-like cysteine protease (TvCPCA1), and purine nucleosine phosphorylase (TvCleoside phosphorylase).**

|  | Receptors |  |  |  |  |  |  |
| --- | --- | --- | --- | --- | --- | --- | --- |
|  | TvCPCAC1 | TvTPI | TvLDH | TvMGL | TvCP2 | TvPNP | TvTrxR |
| Arachidonic acid | -4.5 | -4.4 | -6.7 | -4.9 | -4.2 | -3.2 | -4.8 |
| Cis-8,11,14-Eicosatrienoic acid | -4.1 | -3.4 | -5.5 | -4.9 | -4.2 | -3.6 | -4.2 |
| Eicosapentaenoic acid | -4.2 | -3.8 | -6.6 | -5.3 | -4.5 | -2.8 | -5.4 |
| Heptadecanoic acid | -3 | -3.8 | -4.6 | -3.7 | -4 | -1.9 | -4.3 |
| Lauric acid | -3.1 | -3.7 | -4.8 | -4 | -3.7 | -2 | -4 |
| Linoleic acid | -4.1 | -3.6 | -5.4 | -4.3 | -4.3 | -3 | -5.6 |
| Myristic acid | -3.4 | -2.9 | -4.9 | -4.7 | -3.7 | -2.6 | -3.7 |
| Myristoleic acid | -3.8 | -2.9 | -5.2 | -4.5 | -3.8 | -0.7 | -4.6 |
| Nervonic acid | -3.7 | -3.7 | -5.3 | -4.2 | -4.1 | -3.2 | -4.3 |
| Oleic acid | -3.9 | -3.6 | -4.7 | -4.5 | -4 | -2.7 | -4.4 |
| Palmitic acid | -3.6 | -3.9 | -4.9 | -4.3 | -3.6 | -1.5 | -3.8 |
| Palmitoleic acid | -3.6 | -3.1 | -5.5 | -4.7 | -3.6 | -2.6 | -4.6 |
| Pentadecanoic acid | -3.5 | -3.9 | -4.7 | -3.8 | -3.2 | -3 | -4.2 |
| Stearic acid | -3.4 | 0 | -5 | -4.2 | -4.1 | -2.4 | -5.4 |
| Metronidazole | -4.4 | -5.2 | -6 | -6.5 | -5.5 | -2.9 | -5.5 |

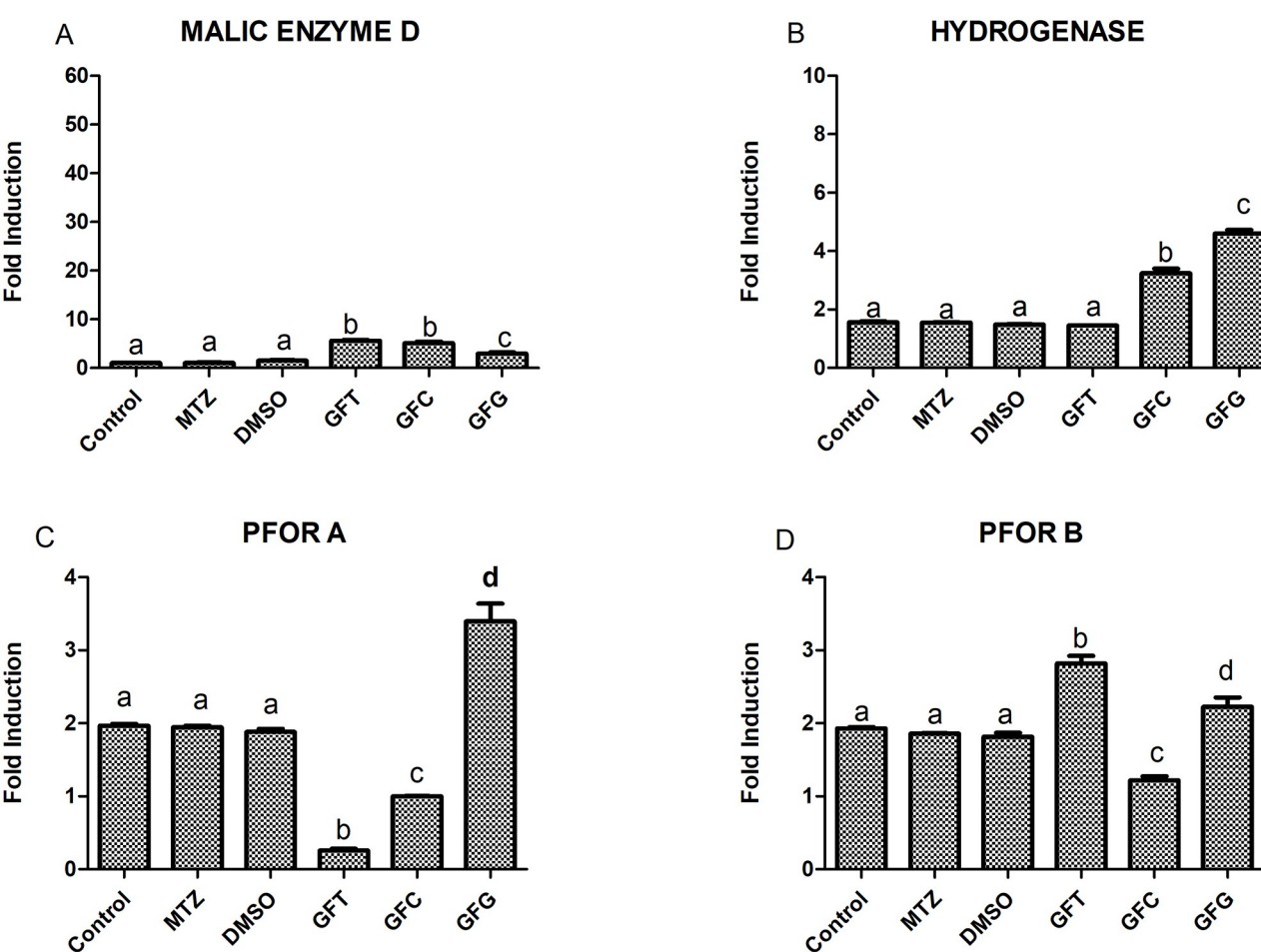

**Fig 5. Gene expression profile of *T. vaginalis* after treatment with the lipid extracts GFG, GFC, and GFT at a concentration of 100 μg/mL.** Gene expression was determined through qRT-PCR. Data are presented as the mean ±standard deviation of a minimum of three independent experimentsControl (negative control—untreated *T. vaginalis*), DMSO (solubilization vehicle), and MTZ (metronidazole).

tracing of the constituent compounds of these extracts. The GC-FID technique is well known in the industry for the separation and identification of fatty acids from oils and fatty formulations with efficiency, accuracy, and accessibility [28]. In the present study, GC-FID was employed to trace a phytochemical profile of the fatty acids constituents of the lipid extracts.

Several of the identified constituent compounds are already reported in the literature for exhibiting varied bioactivities, including antiparasitic effects [29–31]. In this context, the present study was aimed to evaluate the anti-*T. vaginalis* activity of the lipid extracts derived from the macroalga native to the subantarctic regions at its different developmental stages.

The results of the present study provided evidence of the potential antiprotozoal effect of the GFG, GFC, and GFT extracts *in vitro* as the viability of *T. vaginalis* trophozoites was observed to decrease (Fig 1). Furthermore, the *in silico* analysis of the gene expression of this parasite after treatment with the extracts revealed the potential mechanisms of action underlying the role of these extracts as alternatives to the commercially used chemotherapeutics.

Although differences in the MIC values were observed among GFG, GFC, and GFT when evaluated against *T. vaginalis* trophozoites, considerable levels of parasite inhibition were achieved using all extracts. These differences in the inhibitory effects of the extracts could be

attributed to the different developmental stages at which these extracts were derived from the macroalgae. Furthermore, the constituent fatty acids exhibited variability among the extracts, in terms of both concentrations and the presence or absence of the constituents. Despite this, the effect was observed that the evaluated extracts exhibited considerable levels of antiparasitic activity against *T. vaginalis*, and were, therefore, promising candidates for use as antiparasitics. This is also corroborated by the previous studies investigating the use of seaweeds as antiparasitics.

Bonde *et al.* (2021) [32] for example, demonstrated the parasitic development inhibition effect of the extracts derived from the seaweed from the cold waters of the Nordic region. In addition, the presence of fatty acids, including stearidonic acid, alpha-linolenic acid, docosa-hexaenoic acid, arachidonic acid, and eicosapentaenoic acid, was demonstrated to promote anthelmintic activity. Arachidonic acid and eicosapentaenoic acid were detected in all the extracts evaluated in the present study. The bioactive activity of oleic, linoleic, palmitic, and other unsaturated acids has been described as important for protection against microbial infections [33].

Das, (2018) [34] reported that macrophages released certain types of unsaturated fatty acids in the alveolar fluid to execute an antimicrobial action. This phenomenon has also been observed in certain types of leukocytes and T and B lymphocytes, which release the same acids, in certain situations, as the defense response to protect against infections. The results of the present study revealed that unsaturated fatty acids constituted almost 50% of the total lipid composition in each extract, which supports the hypothesis that these fatty acids were respon-sible for the inhibition of *T. vaginalis* trophozoite development.

In addition, many other naturally occurring compounds as well as fatty acids have already shown antiparasitic activity. Castillo *et al.* (2022) [35], presented the antiprotozoal effect of a new flavone, a chemical compound belonging to the flavonoid family, identified in the metha-nol extract of four different plant species, reducing the viability of epimastigote forms of Try-panosoma cruzi by up to 50%. In addition, Veas *et al.* (2020) [36], observed the anti-protozoal activity of phenyl compounds and terpenoids obtained from ethanolic strata Chlamydomonas reinhardtii and Scenedesmus obliquus as possible promising biopharmaceuticals for Chagas disease, thus corroborating the results shown in our study, demonstrating the important role of natural bioactives in combating parasitic diseases.

As novel pharmacological bioproducts continue to be produced, evaluating the cytotoxic effects of their constituent compounds is necessary. In this context, the cytotoxic effects of the extracts against the HMVII cell line and VERO cell line were analyzed in the present study using the MTT assay. The MIC values (100 μg/mL) of GFG, GFC, and GFT did not demon-strate any cytotoxic effects in the VERO cell line, and no significant differences were observed compared to the group of untreated cells. In the HMVII cell line, a decrease of 22.68%, 19.36%, and 17.4% was observed in the growth of cells when using GFG, GFC, and GFT, respectively.

According to the standards established by the International Organization for Standardiza-tion (ISO) in its ISO 10993–5: 2009, reductions in cell viability equal to or greater than 30% in cytotoxic assays should be considered toxic [37] (ISO, 2009: JUNG et al., 2019).

In the present study, this percentage of cell viability reduction was not reached in either of the tested cell lines for any of the extracts, corroborating the feasibility of these extracts as promising drugs for the treatment of trichomoniasis. The molecular docking analysis is con-sidered important for exploring novel molecules that could be used as potential drugs for the treatment of diseases, such as trichomoniasis.

The *in silico*, observation of the interactions of the fatty acids present in the evaluated extracts with important enzyme targets involved in the parasitic mechanism of *T. vaginalis*

suggested that the antiparasitic action mechanism of the extracts could involve these fatty acids. The best scores observed in the present study were those obtained for arachidonic acid, eicosapentaenoic acid, and linoleic acid in interaction with the enzyme active sites TvLDH and TvxR.

The investigation of these targets as the possible routes of action for the antiparasitic effect on *T vaginalis* is of great importance for the development of novel formulations. TvLDH, for example, is important for the glycolysis process, where it functions in the catalytic conversion of lactate into pyruvate, thereby being essential for the survival of the parasite. Since TvLDH is not similar to the LDH in human cells, the former is an interesting target for novel drugs [38], with selectivity in their mechanism of action. Thioredoxin reductase, on the other hand, protects against oxidative damage in the parasite as high concentrations of oxygen could otherwise promote the inactivation of enzymes that are vital to the metabolic pathways in the hydrogenosome.

Thioredoxin reductase is also an important target for the drug MTZ, which reacts with the electrons of this enzyme and thereby induces an entire chain of generating nitro-toxic radicals, which leads to the death of the parasite [34, 35]. Therefore, when interacting with these targets, the extracts evaluated in the present study exhibited a potential for use in the development of a novel drug formulation.

In order to better understand the possible routes of action of the antiparasitic activity of GFG, GFC, and GFT, the gene expressions of the enzymes involved in the metabolism of *T. vaginalis* were observed after 24 h of exposing the trophozoites to these compounds. The changes in the mRNA levels of the investigated enzymes (PFOR A, PFOR B, Malic Enzyme D, and Actin Hydrogenase) after treatment were determined.

The correlations of the inhibition of the growth of the parasite upon treatment with GFG, GFC, and GFT could, therefore, be inferred. It was observed that, after the period of exposure to the extracts, the changes in the mRNA levels of these target enzymes were significantly different from those observed in the control groups (untreated cells, cells treated with the solubilization vehicle, and the group treated with the commercially available drug MTZ for the treatment of trichomoniasis).

The parasite *T. vaginalis* is an uncommon eukaryote as it has hydrogenosomes in place of mitochondria. The metabolism of this parasite involves the action of PFOR and its A and B subunits along with the malic enzyme, which together play several crucial roles. Among these roles, the most important one is the conversion of pyruvate into acetyl-CoA, which serves as a substrate in the synthesis of ATP. In addition, the hydrogenase catalysis reaction is observed in the fermentation process of pyruvate, producing molecular hydrogen [39, 40].

Therefore, it was hypothesized that the changes affecting these enzymes might cause serious issues in the development of the parasite. The presence of hybridogenosome in *T. vaginalis*, which differentiates it from the other organisms that contain mitochondria, has driven the search for drugs exhibiting selectivity for the enzymes present in this organelle of the parasite while not affecting the host cells [41]. In this regard, the GFG, GFC, and GFT extracts emerged as promising candidates.

## Conclusion

The natural extracts GFG, GFC, and GFT derived from subantarctic marine macroalgae *Gigartina skottsbergii* after exhibited significant trichomonicidal activity against *T. vaginalis* trophozoites. Among the three extracts, GFG exhibited the highest inhibition index (100%). The *in silico* analysis through molecular docking revealed that all three extracts could interact with the active sites of the enzymes relevant to the parasitic metabolism, such as TvrxR and TvLDH,

which are already known to be important targets for novel pharmaceutical formulations being developed against this disease. None of the three extracts exhibited significant levels of cytotoxicity in either of the cell lines evaluated. Moreover, alterations observed in the gene expression profile of *T. vaginalis* trophozoites 24 h after exposure to the MIC of the constituent compounds of the three extracts could be related to the antiparasitic effect demonstrated by these extracts. The present study demonstrated that novel bioproducts could be obtained from this macroalgal species for potential promising application in the treatment of the disease caused by *T. vaginalis*.

## Supporting information

**S1 Data.**
(XLSX)

## Author Contributions

**Conceptualization:** Cláudio Martin Pereira de Pereira, Sibele Borsuk.

**Data curation:** Fabiana Kommling Seixas.

**Formal analysis:** Mara Thais de Oliveira Silva, Ângela Sena-Lopes, Frederico Schmitt Kremer.

**Investigation:** Tallyson Nogueira Barbosa, Ângela Sena-Lopes, Fernanda Severo Sabedra Sousa.

**Methodology:** Tallyson Nogueira Barbosa, Mara Thais de Oliveira Silva, Cláudio Martin Pereira de Pereira.

**Project administration:** Sibele Borsuk.

**Software:** Frederico Schmitt Kremer.

**Supervision:** Sibele Borsuk.

**Validation:** Tiago Veiras Collares.

**Visualization:** Fabiana Kommling Seixas.

**Writing – original draft:** Tallyson Nogueira Barbosa, Frederico Schmitt Kremer.

**Writing – review & editing:** Sibele Borsuk.

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
