## [Decision Letter · Decision Letter 0]

31 Jan 2023

PONE-D-22-31808BIOPROSPECTION OF THE TRICHOMONACIDAL ACTIVITY OF LIPID EXTRACTS DERIVED AT DIFFERENT STAGES OF DEVELOPMENT FROM Gigartina skottsbergii AGAINST Trichomonas vaginalisPLOS ONE

Dear Dr. Borsuk,

Thank you for submitting your manuscript to PLOS ONE. After careful consideration, we feel that it has merit but does not fully meet PLOS ONE’s publication criteria as it currently stands. Therefore, we invite you to submit a revised version of the manuscript that addresses the points raised during the review process.

We look forward to receiving your revised manuscript.

Kind regards,

Suprabhat Mukherjee, Ph.D.

Academic Editor

PLOS ONE

Journal Requirements:

2. In your Methods section, please provide additional information regarding the permits you obtained for the work. Please ensure you have included the full name of the authority that approved the field site access and, if no permits were required, a brief statement explaining why

“This study was financed in part by the Coordenação de Aperfeiçoamento de Pessoa de Nível Superior – Brasil (CAPES) – Finance Code 001.”

6. Thank you for stating the following in your Competing Interests section: 

“No”

7. In your Data Availability statement, you have not specified where the minimal data set underlying the results described in your manuscript can be found. PLOS defines a study's minimal data set as the underlying data used to reach the conclusions drawn in the manuscript and any additional data required to replicate the reported study findings in their entirety. All PLOS journals require that the minimal data set be made fully available. For more information about our data policy, please see http://journals.plos.org/plosone/s/data-availability.

Additional Editor Comments:

Authors must improve the discussion by comparing the efficacy of the fatty acid rich extracts with other extracts that are available in the natural source used. Author may follow and cite Doi: 10.2174/1389557516666151120121036.

Reviewers' comments:

Reviewer's Responses to Questions

**Comments to the Author**

1. Is the manuscript technically sound, and do the data support the conclusions?

Reviewer #1: Yes

Reviewer #2: Yes

2. Has the statistical analysis been performed appropriately and rigorously? 

Reviewer #1: Yes

Reviewer #2: N/A

3. Have the authors made all data underlying the findings in their manuscript fully available?

Reviewer #1: Yes

Reviewer #2: Yes

4. Is the manuscript presented in an intelligible fashion and written in standard English?

Reviewer #1: Yes

Reviewer #2: Yes

5. Review Comments to the Author

Reviewer #1: The manuscript entitled “BIOPROSPECTION OF THE TRICHOMONACIDAL ACTIVITY OF LIPID EXTRACTS DERIVED AT DIFFERENT STAGES OF DEVELOPMENT FROM Gigartina skottsbergii AGAINST Trichomonas vaginalis” is a nice piece of work describing the in vitro and in silico anti-parasitic activity of Gigartina skottsbergii against Trichomonas vaginalis. But before it can be accepted for publication in this high quality journal the following issues are needed to be addressed properly

1. Please provide a graphical abstract with brief caption

2. It remains elusive how valid the results from the shown experiments are. How many replicates were performed for each culture condition? So the description and interpretation of the data should be done more carefully.

3. Indicate the numbers of parasites as replicate for each experiment in the methodology section or figure legend.

4. The structural or ultra-structural changes in cell morphology after exposure to Gigartina skottsbergii extract should be studied.

5. Abbreviations in the picture should be explained.

Reviewer #2: Authors tried to established the trichomonacidal activity of fatty acids derived from the marine macroalgae Gigartina skottsbergii . Overall the article is interesting and can be published after some minor revisions as follows:

General Comments:

1. Several abbreviations are used in the text without full form. One time mention of the full form of the used abbreviations is required for general readers

2. Though overall language is good, but there is a scope of improving the standard of language. In some cases it is too wordy and unnecessary!

3. Images/ Graphs can be more attractive and colourful.

Specific Comments

Title:

It is too wordy! It can be shortened like “Bioprospection of the trichomonacidal activity of lipid extracts derived from marine macroalgae Gigartina skottsbergii”

Abstract:

Line 31: Trichomonas vaginalis should be italic.

Line 34, 39: in vitro and in silico should be italic and the pattern should be maintained all through the manuscripts

Line 34-35: ‘marine microalgae’ should be placed before ‘Gigartina skottsbergii’

Introduction:

Line 74: ‘….considered important considering …….’ Rewrite the sentence.

Line 78: Instead of ‘non-viral infections’ write ‘protozoan infection’

Line 79: Add ‘particularly in female’ after ‘genitourinary system’

Material and Methods:

Macroalgae Specimens

Line 93-99: How the specimen is identified? What is its accession no.? Necessary information should be provided

Extraction and Derivatization

Line 103: Why the fatty acids are only extracted? Are there any clues for using fatty acids as anti-protozoan drug? Because alga produces an array of secondary metabolites such as alkaloids, flavonoids, glycosides, terpenoids, and phenazines which have alleged medicinal properties!

Line 103: Bligh and Dyer (B&D) - year and reference no. is missing

Line 110: ‘described by [15]’- author is missing!

Instrumentation and Quantification

Line 130-134: ‘Methanol, chloroform, …………..purchased from Supelco (Bellefonte, USA).’ Unnecessary and delete from the text.

Line 141: Full form of ‘TvMGL, TvPNP, TvTPI, and TvLDH’ should be given

Line 145-46: ‘Linker structures ….. PubChem program’.- unnecessary and delete

Line 149,152: ‘Autodock’ should be ‘AutoDock’

Parasite culture conditions

Line 166: Full form of ‘GFG, GFT, and GF’ should be given

Line 166: Correct the information ‘Sena-Lopes (et al., 2017)’- et al. should be italic and outside the parenthesis, also provide the reference no. within [ x]

Line 173: ‘2.6 × 105’ should be ‘2.6 × 105’

Line 180: ‘CO2’ should be ‘CO2’

Cytotoxicity Assay

Line 203: Navarrete-Vázquez (et al., 203 2015) - et al. should be italic and outside the parenthesis, also provide the reference no. within [ x]

Gene expression analysis of T. vaginalis

Line 237-238: ‘Forward5'-CG….AGG3'andReverse 5'-TCCT…..AAC-3'- provide space in appropriate places!

Results:

Anti-T. vaginalis Activity

Line 277: 89,61% and 86,95%, should be 89.61% and 86.95%, respectively

Molecular docking

Line 331: Required space is missing in ‘(∆Gbinding)’

DISCUSSION

Line 384: Few references are required for the comment “Several of the identified constituent compounds are already reported in the literature…”

Line 398-399: ‘Therefore, it was established that the evaluated extracts exhibited…’ this sentence is not in conformity with the previous sentence. Rewrite accordingly

Line 402: ‘Bonde (et al., 2021)’- et al. should be italic and outside the parenthesis, also provide the reference no. within [ x]

Line 448-449: ‘(LEITSCH; KOLARICH; DUCHÊNE, 2010; JONES et 449 al., 2016)’- it is not the Plos Ones’ reference style!

Conclusion:

Line 478: Write ‘Gigartina skottsbergii’ after ‘ subantarctic marine algae’

Line 479: ‘T. vaginalis’ should be italic

6. PLOS authors have the option to publish the peer review history of their article (what does this mean?). If published, this will include your full peer review and any attached files.

Reviewer #1: **Yes: **Dr. Priya Roy

Reviewer #2: No

---

## [Author Response · Author response to Decision Letter 0]

21 Mar 2023

Journal Requirements:

Response: Thank you for your comment, manuscript adjustments were made.

2. In your Methods section, please provide additional information regarding the permits you obtained for the work. Please ensure you have included the full name of the authority that approved the field site access and, if no permits were required, a brief statement explaining why

Response: Thank you for your comment, manuscript adjustments were made.

4. We note that the grant information you provided in the ‘Funding Information’ and ‘Financial Disclosure’ sections do not match. When you resubmit, please ensure that you provide the correct grant numbers for the awards you received for your study in the ‘Funding Information’ section.

“This study was financed in part by the Coordenação de Aperfeiçoamento de Pessoa de Nível Superior – Brasil (CAPES) – Finance Code 001.”

6. Thank you for stating the following in your Competing Interests section: 

“No”

7. In your Data Availability statement, you have not specified where the minimal data set underlying the results described in your manuscript can be found. PLOS defines a study's minimal data set as the underlying data used to reach the conclusions drawn in the manuscript and any additional data required to replicate the reported study findings in their entirety. All PLOS journals require that the minimal data set be made fully available. For more information about our data policy, please see http://journals.plos.org/plosone/s/data-availability.

Additional Editor Comments:

Authors must improve the discussion by comparing the efficacy of the fatty acid rich extracts with other extracts that are available in the natural source used. Author may follow and cite Doi: 10.2174/1389557516666151120121036.

Response: Thank you for your comment, manuscript adjustments were made in line 422-430.

RBeviewers’ comments:

Reviewer #1: 

The manuscript entitled “BIOPROSPECTION OF THE TRICHOMONACIDAL ACTIVITY OF LIPID EXTRACTS DERIVED AT DIFFERENT STAGES OF DEVELOPMENT FROM Gigartina skottsbergii AGAINST Trichomonas vaginalis” is a nice piece of work describing the in vitro and in silico anti-parasitic activity of Gigartina skottsbergii against Trichomonas vaginalis. But before it can be accepted for publication in this high quality journal the following issues are needed to be addressed properly.

1. Please provide a graphical abstract with brief caption.

Response: Thank you for your comment. Graphical abstract with brief caption was provided.

2. It remains elusive how valid the results from the shown experiments are. How many replicates were performed for each culture condition? So the description and interpretation of the data should be done more carefully.

Response: Thanks for the comment. The number of replicates and repetitions performed in parasite and/or cell culture experiments were described throughout the text.

Lines 174-175 “All assays were performed independently in triplicate.”

3. Indicate the numbers of parasites as replicate for each experiment in the methodology section or figure legend.

Response: We appreciate the comment. The number of parasites used in the study has been included in the text.

Lines 178-181: “The parasites were seeded at an initial density of 2.6 × 105 trophozoites/mL in TYM medium at a final concentration of 150 ul of T. vaginalis trophozoites/well, and then incubated in the presence of GFG, GFC and GFT lipid extracts previously diluted in dimethylsulfoxide (DMSO)”

4. The structural or ultra-structural changes in cell morphology after exposure to Gigartina skottsbergii extract should be studied.

Response: Thanks for the suggestion. We state that changes in cell morphology after exposure to the extract were studied in our manuscript. Before carrying out the MTT reduction assay, the evaluated cells were observed through optical microscopy regarding their morphological characteristics and cellular integrity. These cells remained considerably normal, even after the entire period of exposure to the evaluated compounds. In addition, these observations were corroborated with the performance of the MTT test, since it demonstrated the permanence of cell viability of the groups submitted to the extracts, after the test period.

5. Abbreviations in the picture should be explained.

Response: Thank you for the comment. The abbreviations in the picture were explained in lines: 290-292 “(A) Gigartina skottsbergii stage gametophidic (GFG), (B) Gigartina skottsbergii stage cystocarpic (GFC) and Gigartina skottsbergii stage tetrasporophidic (GFT)”

Reviewer #2:

Authors tried to established the trichomonacidal activity of fatty acids derived from the marine macroalgae Gigartina skottsbergii. Overall the article is interesting and can be published after some minor revisions as follows:

General Comments:

1. Several abbreviations are used in the text without full form. One time mention of the full form of the used abbreviations is required for general readers

2. Though overall language is good, but there is a scope of improving the standard of language. In some cases it is too wordy and unnecessary!

3. Images/ Graphs can be more attractive and colourful. 

Response: Thanks for the comments.

Specific Comments:

Title: 

It is too wordy! It can be shortened like “Bioprospection of the trichomonacidal activity of lipid extracts derived from marine macroalgae Gigartina skottsbergii”

Response: Thanks for the suggestion. The title has been changed to “Bioprospection of the trichomonacidal activity of lipid extracts derived from marine macroalgae Gigartina skottsbergii”

Abstract:

Line 31: Trichomonas vaginalis should be italic.

Response: Thanks for the comment, the change was made in line 30.

Line 34, 39: in vitro and in silico should be italic and the pattern should be maintained all through the manuscripts.

Response: Thanks for the comment, the change was made in line 33.

Line 34-35: ‘marine microalgae’ should be placed before ‘Gigartina skottsbergii’

Response: Thanks for the suggestion, The term ‘marine macroalgae’ was added in line 33.

Introduction:

Line 74: ‘….considered important considering …….’ Rewrite the sentence.

Response: Thanks for the comment. The sentence has been rewritten in line 73.

Line 78: Instead of ‘non-viral infections’ write ‘protozoan infection’

Response: Thanks for the comment. The sentence has been rewritten in line 77.

Line 79: Add ‘particularly in female’ after ‘genitourinary system’

Response: Thanks for the comment. The sentence was rewritten with the addition of the term ‘particularly in feminine’, in line 78. 

Material and Methods:

Macroalgae Specimens

Line 93-99: How the specimen is identified? What is its accession no.? Necessary information should be provided 

Response: Thanks for the comment. Information has been added in line 92-95 and 102-104.

Extraction and Derivatization

Line 103: Why the fatty acids are only extracted? Are there any clues for using fatty acids as anti-protozoan drug? Because alga produces an array of secondary metabolites such as alkaloids, flavonoids, glycosides, terpenoids, and phenazines which have alleged medicinal properties!

Response: Thanks for the comment. Although many metabolites demonstrate significant biological activity, the present study aimed to specifically evaluate the antiparasitic effect of fatty acids. Such compounds from marine sources have already been demonstrated in the literature regarding their bioactive action, such as, lipids isolated from some species of marine sponges, which showed antiprotozoal activity, inhibiting amastigotes and promastigotes of Leishmania species, acting as an inhibitor of the type 1B topoisomerase enzyme, leading to the death of the parasite (MAYER et al., 2021). In addition, studies conducted by Atolani and collaborators in 2019, reported an antiptotozoal action of oleic, linoleic and palmitic acids, present in Polyalthialongifolia oil in contact with Toxoplasma gondii cultures, preventing its development. In turn, the bioactive activity of oleic, linoleic, palmitic acids and other unsaturated acids have already been reported as bioactive potential in protecting against microbial infections (Pineda-Alegría et al., 2020.)

References: Mayer AMS, Guerrero AJ, Rodríguez AD, Taglialatela-Scafati O, Nakamura F, Fusetani N. Marine Pharmacology in 2016-2017: Marine Compounds with Antibacterial, Antidiabetic, Antifungal, Anti-Inflammatory, Antiprotozoal, Antituberculosis and Antiviral Activities; Affecting the Immune and Nervous Systems, and Other Miscellaneous Mechanisms of Action. Mar Drugs. 2021 Jan 21;19(2):49. doi: 10.3390/md19020049. PMID: 33494402; PMCID: PMC7910995.

Atolani O, Areh ET, Oguntoye OS, Zubair MF, Fabiyi OA, Oyegoke RA, Tarigha DE, Adamu N, Adeyemi OS, Kambizi L, Olatunji GA 2019. Chemical composition, antioxidant, anti-lipooxygenase, antimicrobial, anti-parasite and cytotoxic activities of P olyalthia longifolia seed oil. Med. Chem. Res. 28: 515–527.

Pineda-Alegría JA, Sánchez JE, González-Cortazar M, Von Son-De Fernex E, González-Garduño R, Mendoza-De Gives P, et al. In vitro nematocidal activity of commercial fatty acids and β-sitosterol against Haemonchus contortus. J Helminthol. 2020; 5–8. doi:10.1017/S0022149X20000152.

Line 103: Bligh and Dyer (B&D) - year and reference no. is missing.

Response: Thanks for the suggestion. The author and year were included in line 108.

Line 110: ‘described by [15]’- author is missing!

Response: Thanks for the comment. The author was included in line 116.

Instrumentation and Quantification

Line 130-134: ‘Methanol, chloroform, …………..purchased from Supelco (Bellefonte, USA).’ Unnecessary and delete from the text.

Response: Thanks for the comment. The information was taken from the text.

Line 141: Full form of ‘TvMGL, TvPNP, TvTPI, and TvLDH’ should be given 

Response: Thanks for the comment. Changes have been included in line 142-147.

Line 145-46: ‘Linker structures ….. PubChem program’.- unnecessary and delete

Response: Thanks for the comment. The information was taken from the text.

Line 149,152: ‘Autodock’ should be ‘AutoDock’

Response: Thanks for the comment. The change was made in line 152.

Parasite culture conditions

Line 166: Full form of ‘GFG, GFT, and GF’ should be given

Response: Thanks for the comment. The complete form has been provided in lines 169-171.

Line 166: Correct the information ‘Sena-Lopes (et al., 2017)’- et al. should be italic and outside the parenthesis, also provide the reference no. within [ x] 

Response: Thanks for the comment. The information has been corrected in line 171.

Line 173: ‘2.6 × 105’ should be ‘2.6 × 105’

Response: Thanks for the comment. The correction was made in line 178.

Line 180: ‘CO2’ should be ‘CO2’

Response: Thanks for the comment. The correction was made in line 186.

Cytotoxicity Assay

Line 203: Navarrete-Vázquez (et al., 203 2015) - et al. should be italic and outside the parenthesis, also provide the reference no. within [ x] 

Response: Thanks for the comment. Thanks for the comment. The information has been corrected in lines 208-209.

Gene expression analysis of T. vaginalis

Line 237-238: ‘Forward5'-CG….AGG3'andReverse 5'-TCCT…..AAC-3'- provide space in appropriate places!

Response: Thanks for the comment. The correction was made in lines 241-250.

Results: 

Anti-T. vaginalis Activity

Line 277: 89,61% and 86,95%, should be 89.61% and 86.95%, respectively

Response: Thanks for the comment. The correction was made in line 283.

Molecular docking

Line 331: Required space is missing in ‘(∆Gbinding)’

Response: Thanks for the comment. The correction was made in line 338.

DISCUSSION

Line 384: Few references are required for the comment “Several of the identified constituent compounds are already reported in the literature…”

Response: Thanks for the comment. More references have been added to the text in line 389-390.

Line 398-399: ‘Therefore, it was established that the evaluated extracts exhibited…’ this sentence is not in conformity with the previous sentence. Rewrite accordingly

Response: Thanks for the comment. The sentence has been rewritten as requested in lines 404-405.

Line 402: ‘Bonde (et al., 2021)’- et al. should be italic and outside the parenthesis, also provide the reference no. within [ x] 

Response: Thanks for the comment. The information has been corrected in line 408.

Line 448-449: ‘(LEITSCH; KOLARICH; DUCHÊNE, 2010; JONES et 449 al., 2016)’- it is not the Plos Ones’ reference style! 

Response: Thanks for the comment. The reference has been corrected to suit Plos One's style.

Conclusion:

Line 478: Write ‘Gigartina skottsbergii’ after ‘subantarctic marine algae’

Response: Thanks for the comment. The term ‘Gigartina skottsbergii’ has been added as requested in lines 483-484.

Line 479: ‘T. vaginalis’ should be italic 

Response: Thanks for the comment. Term has been corrected as requested in line 492-493.

---

## [Decision Letter · Decision Letter 1]

24 Apr 2023

BIOPROSPECTION OF THE TRICHOMONACIDAL ACTIVITY OF LIPID EXTRACTS DERIVED FROM MARINE MACROALGAE Gigartina skottsbergii

PONE-D-22-31808R1

Dear Dr. Borsuk,

We’re pleased to inform you that your manuscript has been judged scientifically suitable for publication and will be formally accepted for publication once it meets all outstanding technical requirements.

Kind regards,

Suprabhat Mukherjee, Ph.D.

Academic Editor

PLOS ONE

Additional Editor Comments (optional):

Authors have addressed all the comments.

Reviewers' comments:

Reviewer's Responses to Questions

**Comments to the Author**

1. If the authors have adequately addressed your comments raised in a previous round of review and you feel that this manuscript is now acceptable for publication, you may indicate that here to bypass the “Comments to the Author” section, enter your conflict of interest statement in the “Confidential to Editor” section, and submit your "Accept" recommendation.

Reviewer #2: All comments have been addressed

2. Is the manuscript technically sound, and do the data support the conclusions?

Reviewer #2: Yes

3. Has the statistical analysis been performed appropriately and rigorously? 

Reviewer #2: Yes

4. Have the authors made all data underlying the findings in their manuscript fully available?

Reviewer #2: Yes

5. Is the manuscript presented in an intelligible fashion and written in standard English?

Reviewer #2: Yes

6. Review Comments to the Author

Reviewer #2: Authors are successfully able to comply all the questions raised by the reviewers. They should check the ms inconformity with PlosOne guidelines.

7. PLOS authors have the option to publish the peer review history of their article (what does this mean?). If published, this will include your full peer review and any attached files.

Reviewer #2: **Yes: **Biplob Kumar Modak

---

## [Editor Report · Acceptance letter]

28 Apr 2023

PONE-D-22-31808R1 

Bioprospection of the Trichomonacidal activity of lipid extracts derived from Marine Macroalgae *Gigartina skottsbergii*

Dear Dr. Borsuk:

I'm pleased to inform you that your manuscript has been deemed suitable for publication in PLOS ONE. Congratulations! Your manuscript is now with our production department. 

Kind regards, 

on behalf of

Dr. Suprabhat Mukherjee 

Academic Editor

PLOS ONE